ecosystems, environmental science, ecology

coastal development, habitat complexity, human impacts, niche provisioning, remote sensing

**Author for correspondence:**
Peter J. Lawrence
e-mail: p.lawrence@bangor.ac.uk

†Joint second authors.
‡Third author.
¶Present address: School of Natural and Environmental Sciences, Newcastle University, Newcastle-upon-Tyne NE1 8RU, UK.
‖Present address: Biological Sciences, University of Rhode Island, Kingston, RI 02881, USA.

# Artificial shorelines lack natural structural complexity across scales

Peter J. Lawrence[1], Ally J. Evans[2,†], Tim Jackson-Bué[1,†], Paul R. Brooks[3,‡], Tasman P. Crowe[3], Amy E. Dozier[4], Stuart R. Jenkins[1], Pippa J. Moore[2,¶], Gareth J. Williams[1] and Andrew J. Davies[1,‖]

[1]School of Ocean Sciences, Bangor University, Menai Bridge LL59 5AB, UK
[2]Institute of Biological, Environmental and Rural Sciences, Aberystwyth University, Aberystwyth SY23 3DA, UK
[3]Earth Institute and School of Biology and Environmental Science, University College Dublin, Dublin, Ireland
[4]MaREI, the SFI Research Centre for Energy, Climate and Marine, Environmental Research Institute, University College Cork, Ringaskiddy, Ireland

PJL, 0000-0002-9809-0221; AJE, 0000-0002-8935-925X; TJ-B, 0000-0001-7077-2186;
PRB, 0000-0002-8105-0813; TPC, 0000-0001-9583-542X; AED, 0000-0003-4039-1467;
SRJ, 0000-0002-2299-6318; PJM, 0000-0002-9889-2216; GJW, 0000-0001-7837-1619;
AJD, 0000-0002-2087-0885

From microbes to humans, habitat structural complexity plays a direct role in the provision of physical living space, and increased complexity supports higher biodiversity and ecosystem functioning across biomes. Coastal development and the construction of artificial shorelines are altering natural landscapes as humans seek socio-economic benefits and protection from coastal storms, flooding and erosion. In this study, we evaluate how much structural complexity is missing on artificial coastal structures compared to natural rocky shorelines, across a range of spatial scales from 1 mm to 10 s of m, using three remote sensing platforms (handheld camera, terrestrial laser scanner and uncrewed aerial vehicles). Natural shorelines were typically more structurally complex than artificial ones and offered greater variation between locations. However, our results varied depending on the type of artificial structure and the scale at which complexity was measured. Seawalls were deficient at all scales (approx. 20–40% less complex than natural shores), whereas rock armour was deficient at the smallest and largest scales (approx. 20–50%). Our findings reinforce concerns that hardening shorelines with artificial structures simplifies coastlines at organism-relevant scales. Furthermore, we offer much-needed insight into how structures might be modified to more closely capture the complexity of natural rocky shores that support biodiversity.

## 1. Introduction

For centuries, humans have moved, hardened and steepened coastlines in order to build settlements and exploit coastal resources [1,2]. This long-term trend of coastal hardening continues unabated [3,4], usually via the construction of artificial structures including harbours, seawalls and breakwaters. Artificial shorelines offer ecosystem services to humanity such as enhanced storm protection, access to sustenance, transport links and recreation [2]. They can also provide new habitat, shelter and substrate for marine organisms to colonize [5–7]. However, hard artificial structures built along coastlines can negatively impact habitats and species through placement loss [8] and altered connectivity [9,10], and can facilitate the spread of non-native species [11]. Furthermore, the biological communities found on coastal structures are often different and of lower diversity than those found along natural rocky coastlines [12–14].

One of the key mechanisms that may explain the lower biodiversity seen on artificial coastlines is the general lack of structural (topographic) complexity [14,15]. Structural complexity as a concept crosses the fields of remote sensing, geology, geomorphology and mathematics. Here, we use it to refer to the local

numeric variability of a surface. It is this form of physical surface variation that is a key driver of community composition, functioning and diversity across biomes by providing physical spaces and shelter, and ameliorating strong environmental and competitive pressures across a wide range of spatial and temporal scales [16–19]. In intertidal rocky habitats, for example, structural complexity generates habitat features such as crevices, ridges and holes, which offer secure anchor points and refuge from physical stressors and predation [20,21], allowing a diversity of species to coexist under variable environmental conditions. In essence, in the absence of structural complexity, we must expect lower diversity and reduced ecological resilience to negative community-altering extremes such as those caused by human-induced climate change, extreme events and non-native species [22–25].

Given the fundamental importance of structural complexity in intertidal systems, there is growing focus on ecologically inspired engineering design (commonly termed 'eco-engineering' or 'greening the grey') that aims to increase structural complexity and thus the biodiversity value of artificial coastal structures [26,27]. Such interventions include the addition or removal of material to increase surface relief and create habitat-forming topographic features such as pits, rock pools, crevices and ridges [28–30]. Although various wider contextual factors may influence the biological communities that can survive on structures (e.g. disturbance [31] and water quality [32]), experimental evidence suggests interventions that add structural complexity can increase the abundance and diversity of intertidal species [27] that in turn may increase their aesthetic value [33]. There may be scenarios where a lack of colonization is desirable on coastal structures when 'fouling', particularly by non-native species, is considered problematic [34]. Some surface textures can even be designed specifically to reduce colonization [35]. Nevertheless, in scenarios where it is desirable for artificial shorelines to provide surrogate habitats for coastal biodiversity [36], it is vital that consideration is given to enhancing their structural complexity to provide niches for a wide range of species and size classes, to reduce biodiversity loss [37]. While it is generally accepted that artificial structures are more structurally homogeneous than natural rocky shores and provide fewer niches, few studies have explicitly quantified the deficit in complexity between artificial and natural rocky habitats [13,15]. In order to effectively reduce the structural deficit and improve biodiversity outcomes on coastal structures, we need quantitative measurements over scales relevant to the species that inhabit them and the environmental drivers that shape their ecology and physiology [38,39].

In this study, we used multiple remote sensing methodologies to measure surface complexity for two common forms of artificial structure (rock armour and seawalls) and contrasted this with natural rocky shores. We captured fine-scale variation (1–10 mm) via handheld structure-from-motion photography, medium scale (10–50 cm) using a terrestrial laser scanner, and large scale (1–10 m) with structure-from-motion photography from an uncrewed aerial vehicle (UAV) platform. We hypothesized that distinct scale-specific structural differences exist between artificial structures and natural rocky shores, with natural shores providing a more structurally complex (i.e. overall magnitude in variation) and structurally diverse (i.e. variability in scales and types of structure) habitat across all scales. We present detailed estimates of structural complexity, capturing multiple organism-relevant scales and produce quantitative insight into the specific deficits that arise from the artificial hardening of the coastline.

## 2. Methods

### (a) Study sites

We measured the structural complexity of intertidal substrate at 24 sites (12 artificial coastal structures and 12 natural rocky shores) around the coast of Wales, United Kingdom (figure 1; electronic supplementary material, table S1). Of the artificial structures, six were rock armour and six were near-vertical seawalls (see examples in figures 1 and 2). For every artificial structure, we surveyed a nearby natural rocky shore with similar seaward aspect and wave exposure. All surveys were conducted during the summer of 2018 on spring tides, within 2 h of low water.

### (b) Quantifying substrate structural complexity

We used three remote sensing approaches to capture the three-dimensional structural complexity of the artificial and natural habitats at a range of spatial scales relevant to intertidal rocky shore organisms (electronic supplementary material, table S2): fine scale (1–10 mm), medium scale (10–50 cm) and large scale (1–10 m). Within each site, a defined survey area of 120 m in length parallel to the coast, and up to 50 m in width was centred on the mid-shore region and delineated using a handheld GPS.

At the fine scale, three-dimensional structure was measured using close range structure-from-motion photogrammetry. Ten 50 × 50 cm three-dimensional frames (with six control points to scale and define a local coordinate system) were haphazardly placed within each survey area. Frames were placed on near-horizontal surfaces on rock armour and near-vertical surfaces on seawalls; inclination was matched at the natural shore sampled alongside each structure. In the centre of each frame, we positioned a 25 × 25 cm quadrat and cleared the rock surface of the biota using wire brushes, scrapers and cloths to reveal the underlying surface. A total of 20 photographs were taken of each inset quadrat, 16 in a 4 × 4 grid perpendicular to the surface, and a further 4 at oblique angles from each corner. Using the 20 photographs, three-dimensional models were generated using structure-from-motion software (Agisoft Photoscan [40,41]) and scaled using the control points from the calibration frame allowing for geospatial referencing with an average spatial error of 0.01 mm in all axes.

At the medium scale, site structure was measured using terrestrial laser scanning. A tripod-mounted Leica Geosystems HDS ScanStation C10 terrestrial laser scanner was positioned at between four and eight stations, depending on site geometry and size, ensuring a full field of view (360° horizontal, 270° vertical). Up to six spherical targets were distributed around the survey area for spatial referencing, and scanner stations were georeferenced using dGPS with post-processed kinematic corrections. The resulting point clouds had point spacing of 10 cm at 100 m range, and an individual three-dimensional point precision of 6 mm at 50 m georeferenced in OSGB36 coordinates and elevation relative to Ordnance Datum Newlyn.

At the large scale, structural complexity was measured using structure-from-motion photogrammetry from aerial images. UAV surveys were conducted at 21 of the 24 sites; two natural and one artificial site (electronic supplementary material, table S1) could not be surveyed due to weather, access and UAV flight regulatory issues. At each site, at least three UAV missions were completed. Flights were planned to ensure 80% frontal, side and lateral overlap in images, typically from a height of 40 m taken using a 36-megapixel Sony camera (full details outlined in electronic supplementary material, table S3). As per the fine-scale surveys, UAV images were processed with the same structure-from-motion software to reconstruct the three-dimensional geometry at each site. Similar to the medium-scale survey, georeferencing was achieved using post-processed dGPS ground control points to optimize the alignment of images and generate

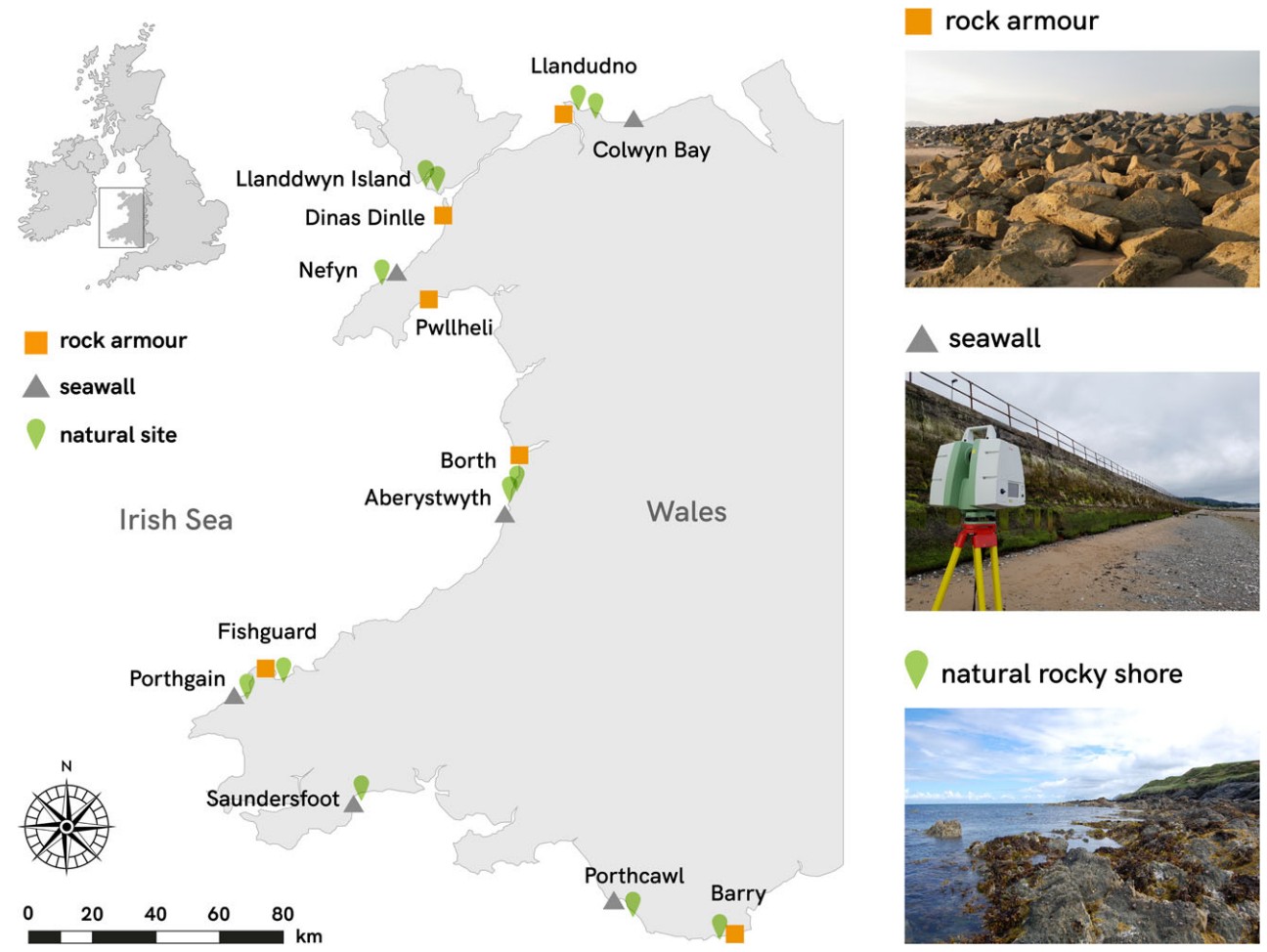

**Figure 1.** Location of study sites, 12 artificial structures (rock armour and seawalls) and 12 natural rocky shores along the coastline of Wales, United Kingdom. For further site details refer to electronic supplementary material, table S1.

scaled georeferenced point clouds achieving an average root mean square error of 41 mm per cloud. All UAV missions were conducted within 1 h either side of spring low tides to capture the maximum aerial exposure of the shorelines sampled.

### (c) Data analysis

#### (i) Point cloud processing

The fine-, medium- and large-scale remote sensing methods generated three-dimensional geometries in the form of point clouds, that were processed, and subsequently analysed using the open-source software CloudCompare [42]. Fine-scale point clouds were clipped to the extent of the cleared $25 \times 25$ cm central quadrat areas. Medium- and large-scale point clouds for each site were clipped to the same survey area bounding polygon ($120 \times 50$ m) to ensure a consistent elevation band and region of data collected across all three survey methods. Areas with soft sediment terrain were removed as they did not contain the hard-substrate habitat of interest to the study.

There are many ways to define structural or habitat complexity. In this study, we calculated the average surface rugosity (the standard deviation of the z-axis in a set two-dimensional ($x$, $y$) space [43]) from all the point clouds of each site. Rugosity is often defined as and is analogous to the 'complexity of a surface'. However, calculations of rugosity can be greatly affected by the slope of a site (i.e. the overall angle of a surface; [44]). We compensated for differing slopes by using rugosity perpendicular to the local surface [43,45], enabling stronger comparison between sites of variable shore inclination. We calculated the average perpendicular rugosity for each site at 12 spatial scales. This was achieved by incrementally increasing the size of the sampling

'window' used to sample the point clouds, between 1 mm (i.e. $1 \times 1$ mm windows) and 10 m (i.e. $10 \times 10$ m windows). The 12 windows for calculating rugosity were distributed across the three categories of scale as follows: fine scale at 1, 5 and 10 mm windows; medium scale at 10, 20, 30, 40 and 50 cm windows; and large scale at 1, 2, 5 and 10 m windows.

The point clouds produced from the three remote sensing methods exhibited some variance in point (observation) density due to the nature of each sensor. To standardize point density between the three methods and improve computing efficiency, all point clouds were randomly thinned to ensure point observations were no closer to each other than a tenth of the smallest window of analysis, i.e. 0.1 mm for fine-scale point clouds (min. window $1 \times 1$ mm), 1 cm for medium-scale point clouds (min. window $10 \times 10$ cm) and 0.1 m for large-scale point clouds (min. window $1 \times 1$ m). The mean and standard deviation of surface rugosity was then calculated per point cloud for each window of resolution to capture within-site average complexity and variability, and the variability between sites. Examples of data outputs in the form of surface rugosity models generated at different scales are shown in figure 2.

#### (ii) Comparing structural complexity across scales

We used a non-parametric bootstrapping approach to determine the median difference in mean surface rugosity between artificial structures (rock armour or seawalls) and their comparative natural rocky shores. We subsequently calculated the uncertainty (95% confidence intervals) based on a total of 1000 bootstrap replicates. Each bootstrap sample was generated via random selection with

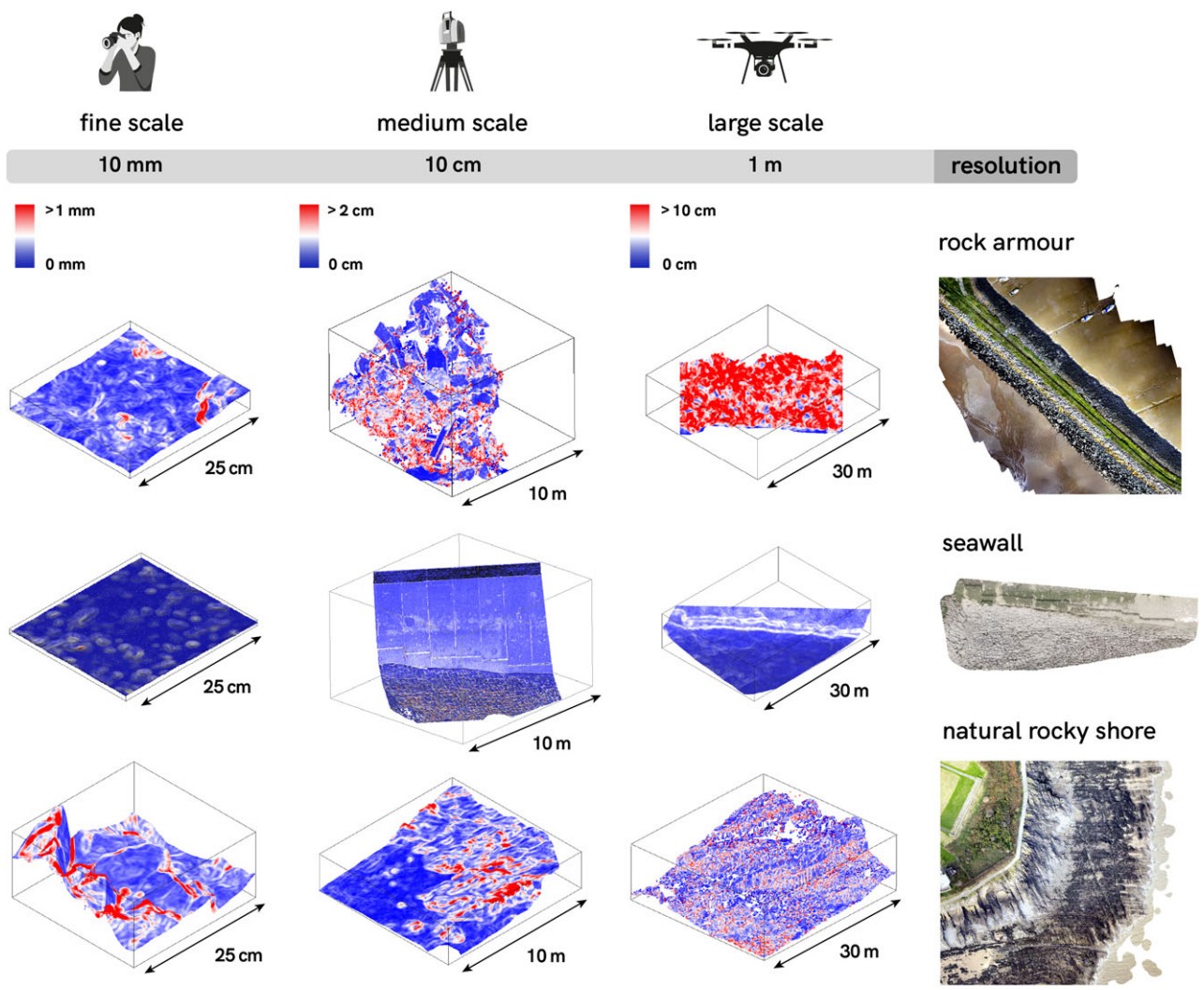

**Figure 2.** Three-dimensional representations of samples of topography recorded from artificial coastal structures (rock armour and seawalls) and natural rocky shores using the three remote sensing methodologies, measured across analysis window scales of 10 mm to 1 m and shown over areas from 25 cm² to 30 m².

replacement for each of the 12 scales measured and consisted of the pooled mean rugosities observed at each type of site (i.e. rock armour, seawall or natural). Non-parametric bootstrapping was selected to avoid assumptions regarding the distribution of likely rugosity, unknown spatial dependence and differing sampling size of the study sites [46–48]. Despite the lower sample size at larger spatial scales (fine scale: 240 quadrats; medium scale: 24 laser scans; large scale: 21 UAV flights), each mean rugosity calculation was obtained from over 1 million observations, providing confidence that the calculated values accurately represented site-level surface rugosity. To compliment the non-parametric bootstrap, we used permutation tests to formally test for differences in the distributions of mean surface rugosity between artificial structures and natural rocky shores [49] across the 12 spatial scales using the 'coin' package in R [50].

### (iii) Within- and between-site variability in structural complexity

Although discrete scales of analysis are easier to interpret, the true surface complexity of a location is the product of complexity measured across all observed scales. To capture this autocorrelation and to compare and contrast the product of all 12 scales of measured complexity between natural shores, rock armour and seawalls, we used non-metric multidimensional scaling (nMDS) based on Gower dissimilarities (as covariates crossed several orders of magnitude and were heterogeneous) using the 'vegan' package in R [51]. We used the standard deviation of complexity rather than the mean in order to compare the variability and predictability of complexity across scales between

sites. To assess significant differences between shoreline types, we used the Hotelling's T² permutation test within the 'Hotelling' package in R [52,53]. This permutational method does not suffer from detection issues when there is high variance within groups and specifically between sample sizes in multivariate comparisons [54].

## 3. Results

The structural complexity of artificial structures was generally lower than that of natural rocky shores (figure 3; electronic supplementary material, table S4). At fine scales, the structural complexity of both rock armour and seawalls was significantly lower than natural shores at all scales (1 mm, 5 mm and 10 mm), with typically 17 to 29% less complexity (figure 3; electronic supplementary material, table S4). At medium scales, the differences in structural complexity between artificial and natural shores were more nuanced. Seawalls were up to 41% less structurally complex than natural shores at all scales (10, 20, 30, 40 and 50 cm). By contrast, the structural complexity of rock armour was more similar to natural shores, with the only significant difference occurring at the 50 cm scale where rock armour was 21% more complex (figure 3; electronic supplementary material, table S4). At larger scales (1, 2, 5 and 10 m), there was higher variability in structural complexity within shore types and fewer significant differences between artificial and natural shores. Seawalls were 38 to 43% less

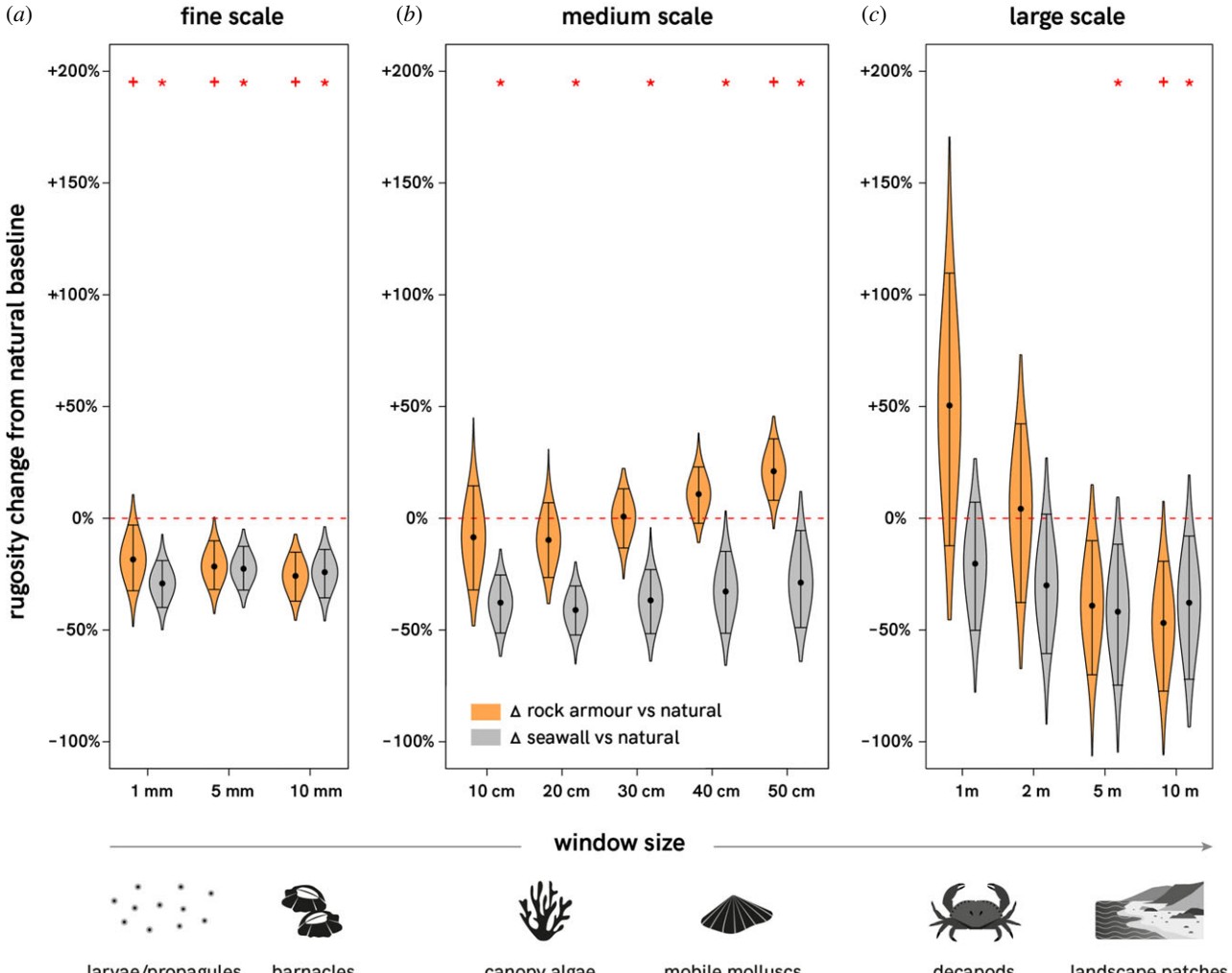

**Figure 3.** The difference in mean surface rugosity of rock armour (orange) and seawalls (grey) compared to the baseline of natural rocky shores (red dashed line) at (*a*) fine, (*b*) medium and (*c*) large scales of measurement. Black dots represent the median difference in mean surface rugosity calculated from 1000 permutations (resampling with replacement). Whiskers represent the upper and lower 95% confidence intervals. Violins plotting above or below the red line suggest the artificial structures have higher or lower rugosity, respectively, than natural shores at each of the given scales (windows). Red crosses and stars indicate significant differences between natural and rock armour, and natural and seawalls respectively (via permutation testing at $p < 0.05$, see electronic supplementary material, table S4). Suggested ecological relevance of different scales is illustrated along the bottom of the x-axis (see electronic supplementary material, table S2 for evidence to support ecological relevance).

complex than natural shores at 5 and 10 m scales, and rock armour was almost 50% less complex at the 10 m scale (figure 3; electronic supplementary material, table S4).

The mean site-level variability of rock armour structural complexity was similar to that of natural shores at two of the three fine scales (1–5 mm), higher than natural shores at three of the five medium scales (30–50 cm) and notably lower than natural shores at three of the four largest scales studied (2–10 m) and the 10 mm scale (figure 4*a*). Mean site-level variability was lower for seawalls than natural shores at all scales, although more so at the finest (1–10 mm) and largest (5–10 m) scales. The site-level overall variability/predictability (represented by the width of the confidence intervals) in rock armour and seawall structural complexity was similar to that of natural rocky shores at fine and medium scales but was considerably lower at the larger scales (1–10 m), especially among rock armour structures. When visualizing all 12 scales of surface rugosity simultaneously, there was some overlap between rock armour structures and natural rocky shores, but these were statistically dissimilar in multiscale complexity ($T^2 = 0.277$, $p < 0.05$). Seawalls were visually and statistically dissimilar from both natural ($T^2 = 1.21$, $p < 0.05$) and rock armour surfaces

($T^2 = -0.72$, $p < 0.05$) (figure 4*b*). Natural shores were also characterized by a more diverse range of site-level rugosities, whereas artificial rock armour and seawalls had more spatially homogeneous structural complexity (figure 4*b*).

## 4. Discussion

As coastlines are developed and artificial structures proliferate, there will be a substantial simplification of structural complexity in coastal habitats over multiple spatial scales. The artificial shoreline structures we surveyed were typically less structurally complex than analogous natural rocky shorelines, but results varied depending on the type of artificial structure and the scale at which complexity was measured. Seawalls were typically approximately 20–40% less complex than natural shorelines at all scales, while rock armour structures were only deficient at the smallest and largest scales measured (approx. 20–50% less complex), but were similar to or more complex than natural shores at medium scales. Both types of artificial structures exhibited high structural similarity between locations, predictably failing to deliver the variation inherent in natural shorelines. While numerous studies have demonstrated that

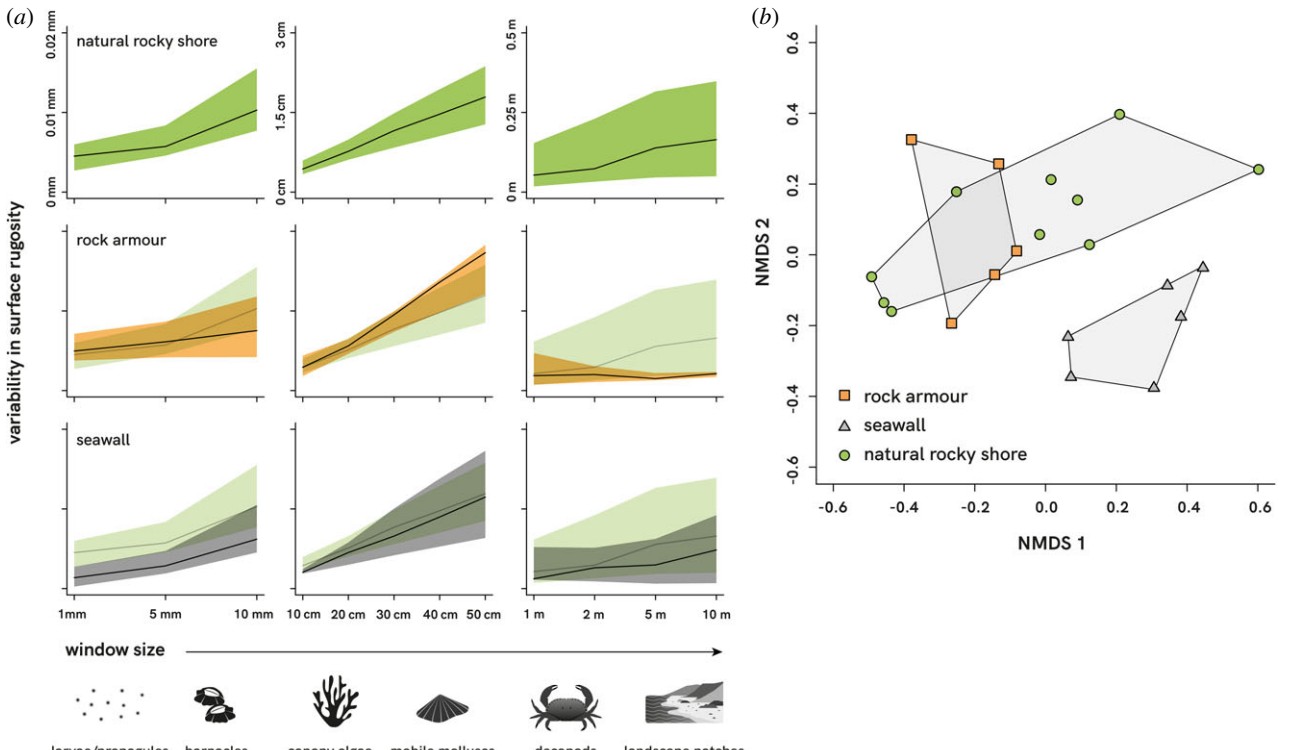

**Figure 4.** (a) Variability (standard deviation) in surface rugosity across scales recorded on natural rocky shores and artificial coastal structures (rock armour and seawalls) around Wales, UK. Solid lines represent the mean site-level variability (50th percentile) and shaded areas indicate the 95th and 5th percentiles. The distribution of variability recorded on natural shores is semi-transparently underlain with those recorded on artificial structures for comparison. (b) nMDS ordination, based on Gower dissimilarities, of variation in multiscale surface rugosity between rock armour (orange squares), seawalls (grey triangles) and natural rocky shores (green circles).

structural eco-engineering interventions on artificial structures can enhance their biodiversity value, either by increasing surface relief or creating habitat-scale features [26,27], a quantitative understanding of the scales at which these interventions are most needed was lacking. Our study identifies the specific scales and amount of structural complexity that is deficient in current rock armour and seawall structure designs when compared to natural shorelines and will inform targeted intervention to reduce the multiscale deficit in topographic complexity between artificial and natural rocky shorelines.

## (a) Potential ecological consequences of multiscale deficits in structural complexity

No matter how we view biodiversity, as intrinsically valuable or as central to human wellbeing and services [55–57], locations with high biodiversity are beneficial to humans and the planet alike [58,59]. With biologically diverse ecosystems at risk globally, there has been an increase in the development of binding targets for their maintenance, protection and restoration. However, for many ecosystems, our mechanistic understanding of how to achieve these targets is lacking [60]. Duarte *et al.* [37] proposed that restoring the three-dimensional complexity of benthic ecosystems should be key to our global efforts to rebuild marine life. In our study, artificial habitats provided a less structurally complex environment that affords less physical space for species colonization, and by extension, narrower niches and fewer microhabitats than natural habitats, a phenomenon not restricted to the coastal environment [61–63]. Such simplification should be of concern due to the explicit linkages between topography, ecological processes and the potential for altered biodiversity patterns [38,64].

Like many before it, our study was predicated upon the central theory that structural complexity is one of the key processes driving biological diversity [65]. Natural structural complexity in the intertidal zone originates through a complex interaction between land geology and the physical forces of the oceans and organisms that act over varying time scales, giving rise to high levels of spatial and temporal heterogeneity. By contrast, artificial structures are created from engineered materials selected for resilience, with designs optimized for coastal protection, or at least to resist erosion and weathering so that they may continue functioning as intended with minimal maintenance requirements. As a result, artificial and natural coastlines represent vastly different environments for organisms. For example, we found that structural features of approximately 1 mm to 1 cm were typically deficient on both rock armour and seawall surfaces. Substrate features at this small scale are known to promote settlement by intertidal organisms, such as barnacles, on natural shorelines [66]. Settled barnacles themselves then create biological structural complexity from their tests that restrict limpet foraging, leading to the reduced top-down control of competitive macroalgae, increased recruitment and growth of mussel beds that in turn reduce thermal pressure and enhance the stability of this community [67–69]. Larger cm- and m-scale complexity would similarly influence niche availability for larger bodied organisms by increasing variation in the carrying capacity of shelters from environmental stressors and predation [70].

## (b) Embedding multiscale structural complexity in engineering

Environmental managers, designers and engineers tasked with developing new infrastructure or retrofitting existing structures

to increase structural complexity for biodiversity enhancement face a challenge. They must balance the usual considerations of cost, engineering and social and environmental impact assessment, while also building-in space in their design envelope for habitat provision [71]. With this balancing act becoming increasingly complex, incorporation of multiscale structural complexity and environmentally sensitive design may be vital for meeting biodiversity targets in the coastal environment [37,72]. Our results highlight the specific scales at which artificial structures are particularly deficient in structural complexity compared to natural shores, and which could be the focus of future targeted interventions. Seawalls were found to be deficient at all scales by approximately 20–40%, while rock armour structures were most deficient at the finest and largest scales (i.e. approximately 20–30% at the mm scale and approximately 40–50% at the 5–10 m scale).

Currently available retrofit or 'bolt-on' eco-engineering designs provide a means of modifying existing structures where deficits in structural complexity were not considered during construction. These are often designed to resemble small- to medium-scale (i.e. cm-scale) habitat features found in natural habitats [27]. For example, drilled or cast pits, grooves and ridges tend to be 1–10 cm in depth, width or height [29,73], while drilled, cast or bolt-on rock pools and holes tend to be 10–50 cm in depth or width [74–76]. Bolt-on rock pool units, by default, add to surface rugosity while simultaneously providing water-retaining refuges. Our results indicate that interventions at these scales can improve the provision of structural complexity on artificial structures, particularly on seawalls where complexity at these scales was consistently deficient. Such interventions are not thought to impact structural integrity [71]. Furthermore, flume experiments have shown that adding topographical complexity to plain seawalls to mimic bolt-on designs can reduce wave overtopping, thus improving their engineering function [77].

Although rock armour was not found to be deficient at medium scales, this does not mean that interventions at this scale are not necessary or valuable for rock armour structures. The high level of complexity in rock armour at the 50 cm to 1 m scale, in particular, reflected the regular rise and fall of uniformly sized and shaped boulders. The void space between boulders may well provide useful habitat [78] but uniform niches that are similar across all rock armour structures will not provide the within or between-site heterogeneity inherent to natural habitats. Furthermore, although water may be retained in sandy recesses between rock armour [79], these do not provide true rock pool habitat [13]. The ecological role played by structural complexity in all studied habitats is clearly more nuanced than the purely objective three-dimensional rugosity measurements we used in this study.

Finer-scale (mm-scale) and larger-scale (m-scale) structural eco-engineering interventions have also been trialled experimentally. Millimetre-scale surface texture can be achieved through material choice [80] or by treating concrete surfaces as they set [16,81]. Metre-scale pre-cast habitat units can be installed in or on structures in the form of blocks or panels [30,73,82]. Interventions at these fine and large scales would be most effective if they were integrated within structures from the outset, rather than bolted-on retrospectively. The largest example of singular coastal eco-engineering intervention units we are aware of are approximately 1.25 m³ BIOBLOCKS [73]. This design creates metre-scale variation in topography due to its size, as well as incorporating smaller multiscale

surface variability that leads to higher species richness than surrounding artificial substrate [73]. Our results suggest that structural complexity remains deficient at scales even larger than those provided by BIOBLOCKS (i.e. greater than or equal to 5 m). Some coastal protection systems comprise modular components at these larger scales (approximately 3 to 5 m [83,84]) and may be readily modified with structural complexity enhancements using specialized formliners or moulds [85]. It may also be possible to make the overall footprint of structures more variable from the outset to increase complexity at these largest scales (i.e. using nonlinear designs), but design modifications at this scale are likely to be driven by cost and engineering requirements and often fall outside the potential remit of eco-engineering approaches. Care should be taken with modifications at a large scale as this may result in increased physical footprint of structures and increased environmental impact associated with such changes. Nevertheless, solutions combining large-scale units with medium-scale habitat features (e.g. rock pools) and fine-scale surface manipulation (e.g. texture or grooves) could provide multiscale structural complexities similar to that of natural shorelines, without increasing structure footprints or compromising their engineering function. Furthermore, directly replicating the full fingerprint of natural reef topography at a variety of spatial scales offers a novel approach to capture a mosaic of structural features that interact to support biodiversity in natural habitats [85].

## 5. Conclusion

We found that the current designs of seawalls and rock amour that are widespread on many developed coastlines provide a poor analogue for natural rocky shorelines in terms of their provision of multiscale structural complexity. Ecologically and in the face of increasing environmental pressures, this lack of complexity represents a considerable deficit in terms of niche provision and is likely to contribute substantially to the lower levels of biodiversity found on artificial structures. From an engineering perspective, there is strong evidence that there is scope for incorporating multiscale surface complexity into the construction of artificial structures with the explicit aim to improve biodiversity outcomes in the next generation of ecologically sensitive design.

Ethics. All UAV flights conformed to UK UAV regulations current at the time of survey (as of June 2018).

Data accessibility. All point clouds, results and processing code developed during this study are available on Zenodo at the following repository: https://doi.org/10.5281/zenodo.4485705.

Authors' contributions. P.J.L.: conceptualization, data curation, formal analysis, investigation, methodology, visualization, writing-original draft, writing-review and editing; A.J.E.: conceptualization, investigation, methodology, writing-original draft, writing-review and editing; T.J-B.: data curation, formal analysis, methodology, writing-review and editing; P.R.B.: conceptualization, methodology, writing-original draft, writing-review and editing; T.P.C.: conceptualization, methodology, writing-review and editing; A.E.D.: data curation, visualization, writing-review and editing; S.R.J.: conceptualization, methodology, writing-review and editing; P.J.M.: conceptualization, investigation, methodology, writing-review and editing; G.W.: formal analysis, methodology, writing-review and editing; A.J.D.: conceptualization, formal analysis, methodology, visualization, writing-original draft, writing-review and editing

All authors gave final approval for publication and agreed to be held accountable for the work performed therein.

Competing interests. We declare we have no competing interests.

**Funding.** This work formed part of the Ecostructure project (www.ecostructureproject.eu), part-funded by the European Regional Development Fund (ERDF) through the Ireland-Wales Cooperation Programme 2014–2020. A.J.D. was supported by the USDA National Institute of Food and Agriculture, Hatch Formula project accession no. 1017848.

**Acknowledgements.** Aerial imagery was provided under contract by RUAS, Southend on Sea, Essex SS1 2PH. We would also like to thank the field crew, especially Harry Thatcher, Hannah Earp and Siobhan Vye, for their perseverance and patience in cleaning and preparing surfaces for three-dimensional analysis.

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
