## [Peer Review File · Proceedings of the Royal Society B: Biological Sciences]

Review History

RSPB-2021-0329.R0 (Original submission)

Review form: Reviewer 1

Recommendation

Accept with minor revision (please list in comments)

Scientific importance: Is the manuscript an original and important contribution to its field?

Excellent

General interest: Is the paper of sufficient general interest?

Good

Quality of the paper: Is the overall quality of the paper suitable?

Excellent

Is the length of the paper justified?

Yes

Should the paper be seen by a specialist statistical reviewer?

No

Do you have any concerns about statistical analyses in this paper? If so, please specify them explicitly in your report.

No

It is a condition of publication that authors make their supporting data, code and materials available - either as supplementary material or hosted in an external repository. Please rate, if applicable, the supporting data on the following criteria.

Is it accessible?

Yes

Is it clear?

Yes

Is it adequate?

Yes

Do you have any ethical concerns with this paper?

No

Comments to the Author

In this study, Lawrence and colleagues use interesting new technologies and good thinking around spatial scales to identify changes in structural complexity between natural and artificial shorelines. The article has nice, clear research aims and hypotheses, and these are well adhered to, and answered, throughout the manuscript. Methods and results are well described and nicely illustrated (figure 4, is particularly interesting and well-illustrated). They conclude with some interesting discussion of how modified structures might be designed differently, or modified, to overcome some of these challenges. The work is very well written with very, very few grammatical errors, and the information flows nicely between sentences and paragraphs. This work is particularly interesting because the question being asked is a relatively simple one (how structurally complex are these ecosystems), but it is tested using nicely implemented new technologies, described and illustrated to make it easy for the reader, and finished with nice conclusions that are applicable across environmental realms and to international audiences. I have only minor comments on this article, which I detail below.

Abstract overall- generally very good, but I was surprised by the lack of quantitative measurements and numbers to support key conclusions given the work undertaken and the description of results in the main body.

Line 41- unclear how these values can be both approximate and a range- is it simply a range? From my reading of the results, I think it is. If there is error associated with that, consider including those values here.

Paragraph commencing line 66- good introduction to the importance of structural complexity, but I was interested in a more thorough definition of this concept. What do you mean by structural complexity in this context? This might seem trivial, but a broader audience might not understand specifically what structural complexity means in the context of hard shorelines. You might also consider introducing the idea here that structural complexity can be measured at multiple scales, and that these scales are crucial in considering the effects of reduced structural complexity for animals.

Line 71- this is an interesting point. Resilience to what, exactly?

Line 120- why approximately? Was it simply the length of the site itself? If so, I'd just be up front about that here.

Line 137- four to eight- again due to size of site? Why the variation here?

Line 154- What processing is involved with these error measurements, and how are they calculated?

Paragraph commencing line 166- just to clarify, these scales here are essentially pixel sizes from which data is extracted from the point cloud?

Line 254- all correct, but the subtlety in your results, particularly for the rock armour vs natural at the medium spatial scales, needs to be mentioned here, as opposed to simply generalising to 'most' spatial scales.

Line 274- 'proposed'

Good discussion of potential interventions and small and medium scales, but I feel that the larger scale is lacking in the current version, especially compared to the other two. Does maximising complexity at this stage principally come down to planning in the design phase, or are there actions that can be implemented retrospectively? How feasible are these sorts of retrospective actions at the larger scales that you're discussing?

Review form: Reviewer 2 (Nathan J. Waltham)

Recommendation

Accept with minor revision (please list in comments)

Scientific importance: Is the manuscript an original and important contribution to its field?

Excellent

General interest: Is the paper of sufficient general interest?

Good

Quality of the paper: Is the overall quality of the paper suitable?

Good

Is the length of the paper justified?

Yes

Should the paper be seen by a specialist statistical reviewer?

No

Do you have any concerns about statistical analyses in this paper? If so, please specify them explicitly in your report.

No

It is a condition of publication that authors make their supporting data, code and materials available - either as supplementary material or hosted in an external repository. Please rate, if applicable, the supporting data on the following criteria.

Is it accessible?

Yes

Is it clear?

Yes

Is it adequate?

Yes

Do you have any ethical concerns with this paper?

No

Comments to the Author

Dear Authors,

This manuscript presents a case study that examines the differences among and between natural

and artificial shoreline habitat complexity. This study is very interesting and moves forward the research domain of eco-engineering, providing a case for scientists to consider habitat complexity in future research projects.

I provide several comments and suggested edits below:

Comments

- Page 4, line 71, “In essence, without structural complexity supporting these processes....., we must expect lower diversity and reduced ecological resilience..... It is therefore vital that where artificial structures are built.....” This seems true, however, diversity might be also linked to environmental conditions or differences (for example temperature) where species diversity are altered due to changes in conditions (e.g. anthropogenic) and not necessary because of changes in habitat complexity. For completeness, the manuscript should at least outline these other processes that are occurring along developing coastlines, which would also influence diversity.
- Following the point above, does diversity include invasive species, which are known to increase on artificial structures (see Tan et al. 2018).
- Page 7 – Methods. Presumably at the rock armour and natural rocky shore sites there would be surface area behind (underneath) the surfaces of rocks/boulders or rocky shoreline that is not represented in the surface elevation methods used here? For example, rock seawalls have voids between each rocks. How important is this to be cognisant of in apply this methodology to other locations? Maybe the method presented here is a standardised procedure to compare complexity among the three different habitat types – measuring only the surface facing complexity of structures.
- Page 16, Line 313, there are also planter box designs that are “bolt on” that are designed to provide a water retaining feature on seawalls to mimic natural rocky shorelines with rock pools – one could argue that these are also designs to increase habitat complexity (Browne and Chapman 2014, Morris et al. 2017), though a recent study in tropical estuaries revealed boxes can present higher water temperature environments which might impact diversity, where water temperature exceeds acute thermal tolerances for species (see Waltham and Sheaves (2020)).
- Overall, while these data are very interesting, it is not clear how or why this study would be a step change for managers that are challenged with approving more coastal development whilst also achieving conservation outcomes. What are the key management implications from this study to consider? Would the authors expect managers to alter the approval design of seawalls etc. to maximise habitat complexity etc. This is not clearly presented in the final conclusion of the manuscript, leaving readers in a position to interpret the data in their own way or local costal ara.

Tan, W.T., Loke, L.H., Yeo, D.C., Tan, S.K. and Todd, P.A., 2018. Do Singapore's seawalls host non-native marine molluscs?. *Aquatic Invasions*, 13(3).

Browne, M. A., and M. Chapman. 2014. Mitigating against the loss of species by adding artificial intertidal pools to existing seawalls. *Marine Ecology Progress Series* 497:119-129.

Morris, R. L., M. G. Chapman, L. B. Firth, and R. A. Coleman. 2017. Increasing habitat complexity on seawalls: Investigating large-and small-scale effects on fish assemblages. *Ecology and Evolution* 7:9567-9579.

Waltham, N., and M. Sheaves. 2020. Thermal exposure risks to mobile tropical marine snails: are eco-engineered rock pools on seawalls scale-specific enough for comprehensive biodiversity outcomes? *Marine pollution bulletin* 156:111237.

Decision letter (RSPB-2021-0329.R0)

09-Apr-2021

Dear Dr Lawrence

I am pleased to inform you that your Review manuscript RSPB-2021-0329 entitled "Artificial shorelines lack natural structural complexity across scales" has been accepted for publication in Proceedings B.

The referee(s) do not recommend any further changes. Therefore, please proof-read your manuscript carefully and upload your final files for publication. Because the schedule for publication is very tight, it is a condition of publication that you submit the revised version of your manuscript within 7 days. If you do not think you will be able to meet this date please let me know immediately.

To upload your manuscript, log into <http://mc.manuscriptcentral.com/prsb> and enter your Author Centre, where you will find your manuscript title listed under "Manuscripts with Decisions." Under "Actions," click on "Create a Revision." Your manuscript number has been appended to denote a revision.

You will be unable to make your revisions on the originally submitted version of the manuscript. Instead, upload a new version through your Author Centre.

1) A text file of the manuscript (doc, txt, rtf or tex), including the references, tables (including captions) and figure captions. Please remove any tracked changes from the text before submission. PDF files are not an accepted format for the "Main Document".

2) A separate electronic file of each figure (tiff, EPS or print-quality PDF preferred). The format should be produced directly from original creation package, or original software format. Please note that PowerPoint files are not accepted.

3) Electronic supplementary material: this should be contained in a separate file from the main text and the file name should contain the author's name and journal name, e.g. `authorname_procb_ESM_figures.pdf`

All supplementary materials accompanying an accepted article will be treated as in their final form. They will be published alongside the paper on the journal website and posted on the online figshare repository. Files on figshare will be made available approximately one week before the accompanying article so that the supplementary material can be attributed a unique DOI. Please see: <https://royalsociety.org/journals/authors/author-guidelines/>

4) Data-Sharing and data citation

It is a condition of publication that data supporting your paper are made available. Data should be made available either in the electronic supplementary material or through an appropriate repository. Details of how to access data should be included in your paper. Please see <https://royalsociety.org/journals/ethics-policies/data-sharing-mining/> for more details.

<http://datadryad.org/submit?journalID=RSPB&manu=RSPB-2021-0329> which will take you to your unique entry in the Dryad repository.

Once again, thank you for submitting your manuscript to Proceedings B and I look forward to receiving your final version. If you have any questions at all, please do not hesitate to get in touch.

Sincerely,
Dr Sasha Dall
mailto:proceedingsb@royalsociety.org

Associate Editor Board Member: 1

Comments to Author:

The authors aim to address a gap to identify changes in structural complexity between natural and artificial shorelines. This manuscript is clearly and concisely written, and the novel results would be of broad interest to the readership of Proceedings B once the minor revisions are addressed.

Both referees find the study interesting and of general importance, with the need for only minor revisions. Reviewer 1 had very minor edits and suggestions to improve the manuscript, including a request for additional literature framing the importance of structural complexity in the Intro section and further reflection on the implications of the study at larger spatial scales in the Discussion section. Reviewer 2 requests further background information in the Discussion that considers how this study would be a step-change for managers that are challenged with approving more coastal development whilst also achieving conservation outcomes. These minor revisions will ultimately improve the manuscript.

Reviewer(s)' Comments to Author:

Referee: 1

Comments to the Author(s)

In this study, Lawrence and colleagues use interesting new technologies and good thinking around spatial scales to identify changes in structural complexity between natural and artificial shorelines. The article has nice, clear research aims and hypotheses, and these are well adhered to, and answered, throughout the manuscript. Methods and results are well described and nicely illustrated (figure 4, is particularly interesting and well-illustrated). They conclude with some interesting discussion of how modified structures might be designed differently, or modified, to overcome some of these challenges. The work is very well written with very, very few grammatical errors, and the information flows nicely between sentences and paragraphs. This work is particular interesting because the question being asked is a relatively simple one (how structurally complex are these ecosystems), but it is tested using nicely implemented new technologies, described and illustrated to make it easy for the reader, and finished with nice conclusions that are applicable across environmental realms and to international audiences. I have only minor comments on this article, which I detail below.

Abstract overall- generally very good, but I was surprised by the lack of quantitative measurements and numbers to support key conclusions given the work undertaken and the description of results in the main body.

Line 41- unclear how these values can be both approximate and a range- is it simply a range? From my reading of the results, I think it is. If there is error associated with that, consider including those values here.

Paragraph commencing line 66- good introduction to the importance of structural complexity, but I was interest in a more thorough definition of this concept. What do you mean by structural complexity in this context? This might seem trivial, but a broader audience might not understand specifically what structural complexity means in the context of hard shorelines. You might also consider introducing the idea here that structural complexity can be measured at multiple scales, and that these scales are crucial in considering the effects of reduced structural complexity for animals.

Line 71- this is an interesting point. Resilience to what, exactly?

Line 120- why approximately? Was it simply the length of the site itself? If so, I'd just be up front about that here.

Line 137- four to eight- again due to size of site? Why the variation here?

Line 154- What processing is involved with these error measurements, and how are they calculated?

Paragraph commencing line 166- just to clarify, these scales here are essentially pixel sizes from which data is extracted from the point cloud?

Line 254- all correct, but the subtlety in your results, particularly for the rock armour vs natural at the medium spatial scales, needs to be mentioned here, as opposed to simply generalising to 'most' spatial scales.

Line 274- 'proposed'

Good discussion of potential interventions and small and medium scales, but I feel that the larger scale is lacking in the current version, especially compared to the other two. Does maximising complexity at this stage principally come down to planning in the design phase, or are there actions that can be implemented retrospectively? How feasible are these sorts of retrospective actions at the larger scales that you're discussing?

Referee: 2

Comments to the Author(s)

Dear Authors,

This manuscript presents a case study that examines the differences among and between natural and artificial shoreline habitat complexity. This study is very interesting and moves forward the research domain of eco-engineering, providing a case for scientists to consider habitat complexity in future research projects.

I provide several comments and suggested edits below:

Comments

- Page 4, line 71, "In essence, without structural complexity supporting these processes....., we must expect lower diversity and reduced ecological resilience..... It is therefore vital that where artificial structures are built...." This seems true, however, diversity might be also linked to environmental conditions or differences (for example temperature) where species diversity are altered due to changes in conditions (e.g. anthropogenic) and not necessary because of changes in habitat complexity. For completeness, the manuscript should at least outline these other processes that are occurring along developing coastlines, which would also influence diversity.
- Following the point above, does diversity include invasive species, which are known to increase on artificial structures (see Tan et al. 2018).
- Page 7 – Methods. Presumably at the rock armour and natural rocky shore sites there would be surface area behind (underneath) the surfaces of rocks/boulders or rocky shoreline that is not represented in the surface elevation methods used here? For example, rock seawalls have voids between each rocks. How important is this to be cognisant of in apply this methodology to other locations? Maybe the method presented here is a standardised procedure to compare complexity among the three different habitat types – measuring only the surface facing complexity of structures.
- Page 16, Line 313, there are also planter box designs that are "bolt on" that are designed to provide a water retaining feature on seawalls to mimic natural rocky shorelines with rock pools – one could argue that these are also designs to increase habitat complexity (Browne and Chapman 2014, Morris et al. 2017), though a recent study in tropical estuaries revealed boxes can present higher water temperature environments which might impact diversity, where water temperature exceeds acute thermal tolerances for species (see Waltham and Sheaves (2020)).
- Overall, while these data are very interesting, it is not clear how or why this study would be a step change for managers that are challenged with approving more coastal development whilst also achieving conservation outcomes. What are the key management implications from this study to consider? Would the authors expect managers to alter the approval design of seawalls etc. to maximise habitat complexity etc. This is not clearly presented in the final conclusion of the manuscript, leaving readers in a position to interpret the data in their own way or local coastal area.

Tan, W.T., Loke, L.H., Yeo, D.C., Tan, S.K. and Todd, P.A., 2018. Do Singapore's seawalls host non-native marine molluscs?. *Aquatic Invasions*, 13(3).

Browne, M. A., and M. Chapman. 2014. Mitigating against the loss of species by adding artificial intertidal pools to existing seawalls. *Marine Ecology Progress Series* 497:119-129.

Morris, R. L., M. G. Chapman, L. B. Firth, and R. A. Coleman. 2017. Increasing habitat complexity on seawalls: Investigating large-and small-scale effects on fish assemblages. *Ecology and Evolution* 7:9567-9579.

Waltham, N., and M. Sheaves. 2020. Thermal exposure risks to mobile tropical marine snails: are eco-engineered rock pools on seawalls scale-specific enough for comprehensive biodiversity outcomes? *Marine pollution bulletin* 156:111237.

Author's Response to Decision Letter for (RSPB-2021-0329.R0)

See Appendix A.

Decision letter (RSPB-2021-0329.R1)

23-Apr-2021

Dear Dr Lawrence

I am pleased to inform you that your manuscript entitled "Artificial shorelines lack natural structural complexity across scales" has been accepted for publication in *Proceedings B*.

Data Accessibility section

Open Access

Paper charges

Sincerely,

Ysgol Gwyddorau Eigion
Prifysgol Bangor
Porthaethwy
Ynys Môn
LL59 5AB

e-bost: sos-enquiries@bangor.ac.uk
www: <http://www.sos.bangor.ac.uk>

PRIFYSGOL
BANGOR
UNIVERSITY

School of Ocean Sciences
Bangor University
Menai Bridge
Anglesey
LL59 5AB

e-mail: sos-enquiries@bangor.ac.uk
www: <http://www.sos.bangor.ac.uk>

Appendix A

April 23rd, 2021

Dear Professor Spencer Barrett and editorial team,

Thank you for considering and providing detailed feedback regarding our research article submission to *Proceedings of the Royal Society B: Artificial shorelines lack natural structural complexity across scales*. Ref 2021-0329. We have addressed the comments of both reviewers in full.

Although not requested, to ensure we appropriately addressed the reviewers' comments. We have provided a point-by-point address with associated line numbers (that refer to the revised manuscript).

We would like to thank both reviewers for their helpful and in-depth reviews and we thank your editorial staff for a quick, well communicated, and enjoyable process towards publication.

Sincerely,

Peter Lawrence

Dr Peter J. Lawrence (lead author, on behalf of all other authors)
School of Ocean Sciences, Bangor University, Anglesey, UK.
Email: p.lawrence@bangor.ac.uk

Response to reviewer comments

Reviewer 1

General

In this study, Lawrence and colleagues use interesting new technologies and good thinking around spatial scales to identify changes in structural complexity between natural and artificial shorelines. The article has nice, clear research aims and hypotheses, and these are well adhered to, and answered, throughout the manuscript. Methods and results are well described and nicely illustrated (figure 4, is particularly interesting and well-illustrated). They conclude with some interesting discussion of how modified structures might be designed differently, or modified, to overcome some of these challenges. The work is very well written with very, very few grammatical errors, and the information flows nicely between sentences and paragraphs. This work is particularly interesting because the question being asked is a relatively simple one (how structurally complex are these ecosystems), but it is tested using nicely implemented new technologies, described and illustrated to make it easy for the reader, and finished with nice conclusions that are applicable across environmental realms and to international audiences. I have only minor comments on this article, which I detail below.

We thank the reviewer for these supportive comments.

Point-by-point.

1. Abstract overall- generally very good, but I was surprised by the lack of quantitative measurements and numbers to support key conclusions given the work undertaken and the description of results in the main body.

We agree that our results warranted further expansion in the abstract and have made the following addition to provide more detailed quantitative measurements.

Edit - Line 35 onward – “However, our results varied depending on the type or artificial shoreline and the spatial scale at which complexity was measured. Seawalls were deficient in structural complexity at all scales (typically ~20-40 % less complex than natural shores), whereas rock armour was only deficient at the smallest and largest scales measured (~20-50 % less complex) but similar to or more complex than natural shores at medium scales (up to 50 % more complex).”.

2. Line 41- unclear how these values can be both approximate and a range- is it simply a range? From my reading of the results, I think it is. If there is error associated with that, consider including those values here.

Line 37 - We use approximately “~” since we have rounded and quoted values to the nearest 10 for ease of describing similar (but not equal) results in the abstract. We present a range of values to reflect the

range in deficiency across the different scales measured. More granularity in our results is shown later in the manuscript.

3. Paragraph commencing line 66- good introduction to the importance of structural complexity, but I was interest in a more thorough definition of this concept. What do you mean by structural complexity in this context? This might seem trivial, but a broader audience might not understand specifically what structural complexity means in the context of hard shorelines. You might also consider introducing the idea here that structural complexity can be measured at multiple scales, and that these scales are crucial in considering the effects of reduced structural complexity for animals.

Thank you for highlighting this, we have made several edits to improve the broader understanding of the concept.

Edit – Line 62 – “Structural complexity as a concept crosses the fields of remote sensing, geology, geomorphology, and mathematics. Here we use it to refer to the local numeric variability of a surface. It is this form of physical surface variation that is...”

Edit - Line 74 – “In intertidal rocky habitats, for example, structural complexity generates habitat features such as crevices, ridges and holes, which offer secure anchor points and refuge from physical stressors and predation (Aguilera et al. 2019, Menge and Lubchenco 1981), allowing a diversity of species to co-exist under variable environmental conditions.”

This point continues into point 4 of this response linking the definition

4. Line 71- this is an interesting point. Resilience to what, exactly?

We have made several additions and incorporated some new examples from the literature: This point follows on and now ties directly to item 3.

Edit – Line 62-75 (in total) - “In essence, in the absence of structural complexity, we must expect lower diversity and reduced ecological resilience to negative community-altering extremes such as those caused by human-induced climate change, extreme events and non-native species (Graham et al. 2015, Kovalenko et al. 2012, Nash et al. 2014, Peterson et al. 1998).”

5. Line 120- why approximately? Was it simply the length of the site itself? If so, I’d just be up front about that here.

We have clarified as follows: Line 125 – “approximately” removed this was measured by tape (defined later) and thus as accurate as feasible over the large-scale field operations we conducted.

6. Line 137- four to eight- again due to size of site? Why the variation here?

We have clarified as follows: Line 132 - “between four and eight stations, depending on site geometry and size, ensuring a full field of view”.

7. Line 154- What processing is involved with these error measurements, and how are they calculated?

We have clarified and made a more explicit distinction between method and error - Line 157 – “achieving an average root mean square error of 41 mm per cloud”.

8. Paragraph commencing line 166- just to clarify, these scales here are essentially pixel sizes from which data is extracted from the point cloud?

To address this comment, we made several edits as follows:

Line 172 onward: We moved a definition of rugosity to earlier in paragraph, as this defines how the indices are calculated and better explains how the pixels in the point clouds are used. We then clarified further the sampling windows: “We calculated the average perpendicular rugosity for each site at 12 spatial scales. This was achieved by incrementally increasing the size of the sampling ‘window’ used to sample the point clouds, between 1 mm (i.e., 1 x 1 mm windows) and 10 m (i.e., 10 x 10 m windows)”.

Further references to specific window sizes were added the remainder of the sentence to help formalise our use of scales as different windows of analysis.

9. Line 254- all correct, but the subtlety in your results, particularly for the rock armour vs natural at the medium spatial scales, needs to be mentioned here, as opposed to simply generalising to ‘most’ spatial scales.

We agree and now also refer to this in the abstract (see response to comment 1 above): Edit made – Line 263 – “...but results varied depending on the type of artificial structure and the scale at which complexity was measured. Seawalls were typically ~20-40% less complex than natural shorelines at all scales, while rock armour structures were only deficient at the smallest and largest scales measured (~20-50% less complex), but were similar to or more complex than natural shores at medium scales.”

10. Good discussion of potential interventions and small and medium scales, but I feel that the larger scale is lacking in the current version, especially compared to the other two. Does maximising complexity at this stage principally come down to planning in the design phase, or are there actions that can be implemented retrospectively? How feasible are these sorts of retrospective actions at the larger scales that you’re discussing?

We thank the reviewer for these comments. We agree that some further expansion on scale specific outcomes would make our conclusions stronger, but we must point out the fact that interventions at the larger scales observed are currently rare and one aim of our manuscript is to draw light to this fact.

Reviewer 2 also provided some specific comments in point 15. We provide our response to these comments there.

Reviewer 2

General

This manuscript presents a case study that examines the differences among and between natural and artificial shoreline habitat complexity. This study is very interesting and moves forward the research domain of eco-engineering, providing a case for scientists to consider habitat complexity in future research projects.

We thank the reviewer for these supportive comments.

Point-by-point.

11. Page 4, line 71, “In essence, without structural complexity supporting these processes....., we must expect lower diversity and reduced ecological resilience..... It is therefore vital that where artificial structures are built.....” This seems true, however, diversity might be also linked to environmental conditions or differences (for example temperature) where species diversity are altered due to changes in conditions (e.g. anthropogenic) and not necessary because of changes in habitat complexity. For completeness, the manuscript should at least outline these other processes that are occurring along developing coastlines, which would also influence diversity.

Edit – Line 68 (though to 87) - Structural complexity can act both independently and be embedded within numerous other drivers of biodiversity. We have improved the text with some specific examples:

“In intertidal rocky habitats, for example, structural complexity generates habitat features such as crevices, ridges and holes, which offer secure anchor points and refuge from physical stressors and predation (Aguilera et al. 2019, Menge and Lubchenco 1981), allowing a diversity of species to co-exist under variable environmental conditions. In essence, without structural complexity (Richardson et al. 2017, Sullivan et al. 2017, Weiss et al. 2015), we must expect lower diversity and reduced ecological resilience to negative community-altering extremes such as those caused by human-induced climate change, extreme events and non-native species (Graham et al. 2015, Kovalenko et al. 2012, Nash et al. 2014, Peterson et al. 1998).”

“Although various wider contextual factors may influence the biological communities that can survive on structures (e.g., disturbance: Airoidi and Bulleri 2011; water quality: Perrett et al. 2006)..”

12. Following the point above, does diversity include invasive species, which are known to increase on artificial structures (see Tan et al. 2018).

We added the following text to address this comment:

Edit – Line 87 onward – “There may be scenarios where a lack of colonisation is desirable on coastal structures when ‘fouling’, particularly by non-native species, is considered problematic (Hanson and Bell 1976). Some surface textures can even be designed specifically to reduce colonisation (Scardino and De Nys 2011). Nevertheless, in scenarios where it is desirable for artificial shorelines to provide surrogate habitats for coastal biodiversity (Evans et al. 2019), it is vital that consideration is given to enhancing their structural complexity to provide niches for a wide range of species and size classes, to reduce biodiversity loss (Duarte et al. 2020).”

13. Page 7 – Methods. Presumably at the rock armour and natural rocky shore sites there would be surface area behind (underneath) the surfaces of rocks/boulders or rocky shoreline that is not represented in the surface elevation methods used here? For example, rock seawalls have voids between each rocks. How important is this to be cognisant of in apply this methodology to other locations? Maybe the method presented here is a standardised procedure to compare complexity among the three different habitat types – measuring only the surface facing complexity of structures.

We agree with the reviewer and we discussed the fact that this is a key limitation of all current remote sensing technology that can cover the studied observational scales in our original discussion. See below for the current text provided:

“The high level of complexity in rock armour at the 50 cm to 1 m scale, in particular, reflected the regular rise and fall of uniformly sized and shaped boulders. The void space between boulders may well provide useful habitat (Sherrard et al. 2016) but uniform niches that are similar across all rock armour structures will not provide the within or between-site heterogeneity inherent to natural habitats. Furthermore, although water may be retained in sandy recesses between rock armour (Pinn et al. 2005), these do not provide true rock pool habitat (Firth et al. 2013). The ecological role played by structural complexity in all studied habitats is clearly more nuanced than the purely objective three-dimensional rugosity measurements we used in this study.”

14. Page 16, Line 313, there are also planter box designs that are “bolt on” that are designed to provide a water retaining feature on seawalls to mimic natural rocky shorelines with rock pools – one could argue that these are also designs to increase habitat complexity (Browne and Chapman 2014, Morris et al. 2017), though a recent study in tropical estuaries revealed boxes can present higher water temperature environments which might impact diversity, where water temperature exceeds acute thermal tolerances for species (see Waltham and Sheaves (2020)).

We welcome the reviewer's suggestion and detailed description of bolt on rock pool designs and their value. However, we feel that adding further detailed text will distract the reader from the core narrative of the manuscript.

Edit – Line 331 – added Morris et al. 2017 to reference list.

15. Overall, while these data are very interesting, it is not clear how or why this study would be a step change for managers that are challenged with approving more coastal development whilst also achieving conservation outcomes. What are the key management implications from this study to consider? Would the authors expect managers to alter the approval design of seawalls etc. to maximise habitat complexity etc. This is not clearly presented in the final conclusion of the manuscript, leaving readers in a position to interpret the data in their own way or local costal ara.

Thank you for this this insightful comment, to address this we have added some additional text to the discussion to highlight how our findings can help guide management in improving the design of more ecologically sensitive structures. The added text helps to focus the narrative towards our final concluding statement which already focuses on the management implications of our study without distracting from our core aim, to provide objective science that asks fundamental questions about the nature of artificial structures in the marine environment.

Added to discussion Lines 364 onwards: “It may also be possible to make the overall footprint of structures more variable from the outset to increase complexity at these largest scales (i.e., using non-linear designs), but design modifications at this scale are likely to be driven by cost and engineering requirements and often fall outside the potential remit of eco-engineering approaches. Care should be taken with modifications at the large scale as this may result in increased physical footprint of structures and increased environmental impact associated with such changes. Nevertheless, solutions combining large-scale units with medium-scale habitat features (e.g., rock pools) and fine-scale surface manipulation (e.g., texture or grooves) could provide multiscale structural complexities similar to that of natural shorelines, without increasing structure footprints or compromising their engineering function.”